# Market, power, gift, and concession economies: Comparison using four-mode primitive network models

Takeshi Kato[1]*, Junichi Miyakoshi[2], Misa Owa[2], Ryuji Mine[2]

**1** Hitachi Kyoto University Laboratory, Kyoto University, Kyoto, Japan, **2** Digital Innovation R&D, Research & Development Group, Hitachi, Ltd., Tokyo, Japan

* kato.takeshi.3u@kyoto-u.ac.jp

## Abstract

Reducing wealth inequality is a global challenge, and the problems of capitalism stem from the enclosure of the commons and the breakdown of the community. According to previous studies by Polanyi, Karatani, and Graeber, economic modes can be divided into capitalist market economy (enclosure and exchange), power economy (de-enclosure and redistribution), gift economy (obligation to return and reciprocity), and concession economy (de-obligation to return). The concession economy reflects Graeber's baseline communism (from each according to their abilities, to each according to their needs) and Deguchi's We-turn philosophy (the "I" as an individual has a "fundamental incapability" and the subject of physical action, responsibility, and freedom is "We" as a multi-agent system, including the "I"). In this study, we constructed novel network models for these four modes and compared their properties (cluster coefficient, graph density, reciprocity, assortativity, centrality, and Gini coefficient). Our calculation results show that the market economy has more inequality; the power economy mitigates inequality but cannot eliminate it; the gift and concession economies lead to a more healthy and equal economy; and the concession economy, free from the constraints of obligation to return, is possible without guaranteeing reciprocity. In comparison with the established models, we found that the power economy is equivalent to Barabási–Albert model with power law, that the gift and concession economies can be positioned within variations of Watts–Strogatz model with small-world property, and that the approach from concession to gift economies is more effective in reducing inequality, fostering natural reciprocity while avoiding the constraints of return. We intend to promote the transformation from a capitalist economy to a concession economy through activities that disseminate baseline communism and the We-turn philosophy that promotes concession, that is, developing a cooperative platform to support concession through information technology and empirical research through fieldwork.

**Data availability statement:** All relevant data are within the manuscript and its Supporting Information files.

**Funding:** The author(s) received no specific funding for this work.

**Competing interests:** The authors have declared that no competing interests exist.

## Introduction

Wealth inequality is a significant social problem globally. According to the World Inequality Report 2022, the top 10% of the wealthy account for 76% of the world's wealth, and the top 1% account for 38% [1]. The traditional "elephant curve" is now being replaced by the "Loch Ness monster curve" [1,2]. In the elephant curve, the growth rates of the middle class in emerging countries and the wealthy in developed countries were high, whereas that of the middle class in developed countries was negative. In the Loch Ness monster curve, the growth rate of the middle class in emerging economies is declining, the growth rate of the super-wealthy in developed countries is the only one that is high, and inequality continues to increase.

The capitalist economy has grown through the enclosure of the commons, which leads to the enslavement of labor and the colonization of resources [3–5]. Thus, wealth inequality can be viewed as a consequence of capitalism. In other words, GDP growth contributes to the wealth of capitalists and executives but continues to increase the consumption of resources and energy. Growing inequality reduces well-being, and resource consumption causes environmental problems. To reduce wealth inequality and resource consumption as well as improve well-being and restore the natural environment, an alternative to the capitalist economy is required.

As modified versions of the current capitalist system, economist Freeman's stakeholder capitalism [6], economist Stiglitz's progressive capitalism [7], and philosopher Gabriel's ethical capitalism [8] have been proposed. Alternatives to the capitalist system have also been proposed: the moral economy by economist Bowles [9] and the community economy by economist Rajan [10] and public-policy scholar Hiroi [11]. What these generally have in common are income restrictions and progressive taxation for the wealthy, correcting inequality through stock redistribution, investing in public goods (commons), and revitalizing local communities.

Economist Polanyi has identified three modes of economic classification: reciprocity, redistribution, and market exchange [12]. Furthermore, philosopher Karatani identified four modes of exchange: reciprocity, plunder and redistribution, commodity exchange, and the recovery of reciprocity in a higher dimension [13], while anthropologist Graeber identified three moral principles: hierarchy, exchange, and baseline communism [14]. These can be classified as (i) a gift economy involving reciprocity and the obligation to return, (ii) a power economy involving taxation and redistribution, (iii) a market economy based on the impersonal exchange of goods and money, and (iv) a concession economy that sublimates the gift economy without the obligation to return.

(i) A gift economy is a pattern in which economic agents in an equal relationship exchange goods and services and repeat a cycle of gift and return. A bidirectional relationship is formed between the agents, and an obligation to return is required to maintain an equal relationship between them.

(ii) A power economy is a pattern in which central power gathers wealth and redistributes it from the center to agents at the periphery. Relationships toward the center and relationships from the center to the periphery are formed. Taxes and welfare systems correspond to this mode of economy.

(iii) A market economy is a pattern in which agents exchange goods and money with each other. An equivalent and bidirectional relationship is formed between the agents, but human relationships and obligations such as in a gift economy do not arise.

(iv) A concession economy is a pattern in which agents provide goods and services to other agents without expecting anything in return. The term "concession" is used to distinguish it from the reciprocity of gifts and is related to the "Hōtoku" philosophy (transfer to descendants, others, and society) [15]. A unidirectional relationship is formed from one agent to another, and there is no obligation to return as in a gift economy or equivalence as in a market economy.

Although these four economic modes are representative patterns, they are not mutually exclusive, and can shift between modes or overlap. For example, (i) in a gift economy, if the obligation to return is not fulfilled, equal relationships will transition to hierarchical relationships in (ii) a power economy. Moreover, a baseline communism, which does not require reciprocity or equivalence as well as in (iv) a concession economy, exists as the foundation of all human relationships [14]. However, in this study, we treat these four modes separately to clarify the differences between them.

The current capitalist economy is a combination of (ii) the power economy and (iii) the market economy. In a capitalist economy, the enclosure of commons can be understood as the narrowing of mutual relationships between agents in (iii) a market economy, while redistribution can be understood as the relaxation of this narrowing by the central power in (ii) a power economy. As a solution to the capitalist economy which brings about wealth inequality that cannot be resolved even through redistribution, Karatani and Graeber advocate transforming it into (iv) the concession economy.

In (iv) a concession economy, recovering reciprocity in a higher dimension means suppressing the negative aspects of the obligation to reciprocate gifts and the constraints of the community and promoting the positive aspects of individual freedom. As philosopher Sarthou-Lajus says, it involves rethinking reciprocation as a debt owed not to the immediate recipient but to others and future generations [16]. Baseline communism is a human relationship in which "from each according to their abilities, to each according to their needs," and it can be an alternative to the individualism driving capitalism. Monetary exchange in capitalist markets cuts off social relationships, but baseline communism connects them. The evils of capitalism lie in its emphasis on individualism and economic values, and there is a need to shift towards community and social moral values. Although (iv) a concession economy is considered to be more desirable, whether an actual economy can be established without assuming exchange or reciprocity is a question that needs to be addressed.

Deguchi proposed the We-turn philosophy as a shift in values [17]. The We-turn is an argument that the "I" as an individual is unable to perform physical actions alone because it has a "fundamental incapability" and that the subject of physical action, responsibility, and freedom is turned to "We" as a multi-agent system, including the "I." Owing to our "fundamental incapability," "We" are supported by various people and nature, even without being conscious of reciprocity or return. Under the We-turn, each individual will recognize each other's "fundamental incapability" and diverse values, form a community of shared destiny as co-adventurers, including humans, nature, and the future, and sublimate society into solidarity. In a We-turn society, if physical actions are considered labor, collective action becomes solidarity and cooperation. The We-turn society can be interpreted as the "We economy" [18].

Regarding the economic perspective of the We-turn society, Deguchi states the following in his book [19].

• Good We: Fellowship, equality, hollowness without a power center, cooperativity, voluntary participation, and softened We.

• Bad We: Totalitarianism, exclusivism toward the outside, peer pressure toward the inside, and hardened We.

• Good We are co-adventurers who participate while accepting the risk together.

• We members qualify as co-adventurers (risk-takers) because they are vulnerable and frangible.

• Co-adventurers in the same boat may have economic class divisions, but they are on equal footing as communities of destiny.

 

- Co-adventurers are weighted according to their moral responsibility, but weighting is only a quantitative differentiation between the received profit and associated risk.

- The risk-return allocation is increased or decreased but not monopolized, and everyone receives a return for the risk taken.

To summarize Deguchi's discourse, two things are important in the "We economy:" moral responsibility and risk vulnerability. The We economy makes co-adventurers undertake risk (payment) by their moral responsibility and allocate returns in consideration of their responsibility and risk vulnerability. By considering risk vulnerability and moral responsibility, future generations and the socially vulnerable are prioritized. In other words, the We economy does not necessarily require anything in recompense, as a return in (i) a gift economy or (iii) an exchange in a market economy. The We economy is connected to Karatani and Graeber's (iv) concession economy.

Based on the above, this study aims to clarify the differences in the properties of the four economic modes of (i) the gift economy, (ii) the power economy, (iii) the market economy, and (iv) the concession economy to answer questions about the feasibility of (iv) the concession economy and to provide guidelines for transformation towards that end. The problems of capitalism stem from the enclosure of commons and the breakdown of communities, and their solutions require the regeneration of communities, baseline communism, and the social relationships of "We." Therefore, we attempt to simulate the four economic modes using novel primitive network models that express social relationships and compare their foundational properties.

Numerous studies have analyzed economies using network models. For example, one study used a directed graph network model to explain the mechanisms and inequality of gift economy [20]; another used an exponential random graph model to analyze self-organization and similarity in interregional economic development competition [21]; another used an exponential random graph model to analyze reciprocity in the interbank market [22]; and yet another used an exponential random graph model to analyze structural, economic, geographical, political, and cultural factors in global trade networks [23]. However, none of the previous studies have compared the properties of the four-mode economic models simultaneously.

Representative network models include, for example, the simple random graph model (a graph in which edges are generated randomly), the Erdős–Rényi model (a random graph in which the probability of edge generation follows a binomial distribution) [24], the exponential random graph model (a statistical model with exponential parameters for analyzing network data) [25], the stochastic block model (a random graph in which nodes are divided into communities and the probability of edge generation differs between inside and outside the communities) [26], the Watts–Strogatz model (another random graph in which a regular graph is generated and its edges are rewired based on a predetermined probability) [27], the Barabási–Albert model (another random graph in which nodes of a predetermined degree are added at each step and the edges are connected with a probability proportional to the degree of the existing nodes) [28]. The Watts–Strogatz model exhibits a small-world property where the mean distance between nodes is short relative to the number of nodes, and is often used to simulate communities and acquaintanceships in real society. The Barabási–Albert model exhibits a scale-free property where the degree distribution follows a power law, and is often used to simulate uneven distributions of wealth and population in real society.

This study's purpose is not to examine the properties of random graphs for a given probability or to fit them to measured data but to examine the generation process and properties of the four economic modes. Therefore, we use the simplest random graph model without pre-assigned probabilities or properties. This model is known as the button and thread model and was presented by theoretical biologist Kauffman as an example of self-organization [29]. When the ratio of the number of threads to the number of buttons exceeds 0.5, large clusters suddenly appear, and a phase transition occurs. This is useful in observing the emergence of the four economic modes.

Therefore, in this study, we simulate four economic modes using a simple random graph model by randomly generating edges while changing the list of nodes and the direction of edges. The node list represents the enclosure of (iii) market

economy and the equality of (i) gift and (iv) concession economies, whereas the edge direction represents the exchange in (iii) market economy, the obligation to return in (i) a gift economy, and the de-obligation to return in (iv) concession economy. The redistribution of (ii) the power economy is a mitigation of enclosure; that is, it is positioned as an intermediate between (iii) the market economy and (i) the gift economy.

The remainder of the paper is structured as follows: in the Methods section, we present the calculation methods for the network models corresponding to the four economic modes. Additionally, the Watts–Strogatz model and Barabási–Albert model, which are representative random graph models with structural , i.e.,s, are referred to as comparative models for the above simple random graph models. In the Results section, we present the calculation results of representative properties such as the clustering coefficient, graph density, reciprocity, assortativity, and centrality, including the Gini coefficient, representing the degree of inequality. In the Discussion section, we review the properties of the four economic modes and discuss the challenges and future developments (iv) concession economy.

## Methods

### Network models

In a simple random graph model, first, $n$ nodes are set; then, two nodes are randomly selected from a list of $n$ nodes at each calculation step, and an edge is generated between them [29]. For example, Fig 1 shows the calculation results for the Kauffman button and thread model. The results are for $n = 100$ and three calculations were performed. When the ratio of the number of threads to the number of buttons exceeds 0.5 on the horizontal axis, the largest cluster size (number of nodes) suddenly increases on the vertical axis.

In the network model for four economic modes, based on a simple random graph model, nodes are regarded as economic agents, and edges are regarded as economic relationships for the movement of goods and services. Furthermore, between the four economic modes, the settings for the node list and edge direction were changed. Table 1 shows the four economic modes proposed by Polanyi, Karatani, Graeber, and Deguchi and the model settings corresponding to each economic mode. The following order has been rearranged from the Introduction (i) gift economy, (ii) power economy, (iii) market economy, and (iv) concession economy to (a) market economy, (b) power economy, (c) gift economy, and (d) concession economy to make it easier to observe the correspondence with the model.

Node list: In the four economic modes, the same number of nodes is set at the beginning of the calculation. The initial node list contains all nodes. In (a), the market economy, to simulate enclosure, two selected nodes were added to the node list at each step, and two unselected nodes were deleted randomly. In (b), the power economy, two selected nodes

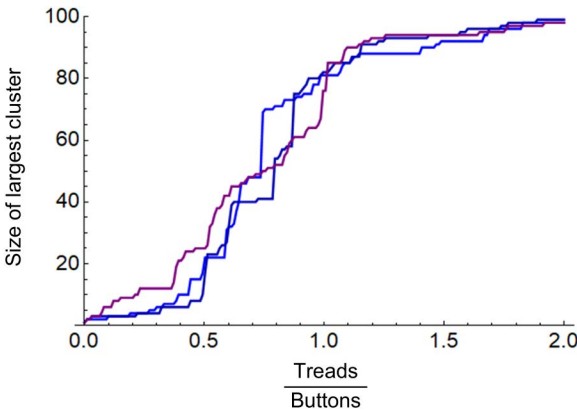

**Fig 1. Calculation results for buttons and threads model.**

were added at each step to simulate the mitigation of the enclosure. In the (c) gift and (d) concession economies, all nodes were listed equally, regardless of the step.

Edge direction: Since the exchange in (a) the market and (b) power economies, and the reciprocity with return in (c) the gift economy are performed, bidirectional edges (two directed edges) are generated between two nodes. In (d) the concession economy, since there is no obligation to return, a unidirectional edge (one-directed edge) is generated between two nodes.

Edge generation: In (a) market, (b) power, and (c) gift economies, two nodes are randomly selected from the node list at each step, and two directed edges are generated between those nodes. In (d) the concession economy, to match the edge generation speed with other economic modes, two nodes were randomly selected two times at each step, and one directed edge was generated between two nodes.

Edge deletion: If we continued to generate edges, all economic modes would approach a complete graph, and the differences in their properties would disappear. Therefore, a mean degree (or total degree = number of nodes × mean degree) was set, and an edge was deleted if the mean degree of the graph exceeds the set degree. The failure of exchange relationship was simulated in (a) the market and (b) power economies, and the failure of reciprocity relationship was simulated in (c) the gift economy, and the edges were deleted randomly. In (d) the concession economy, since there is no mutual relationship inherently, oblivion of unidirectional relationship is simulated over time, and the oldest edges were deleted in order. That is, when a new edge is generated, the oldest edge is deleted.

Based on the above, Fig 2 shows the calculation flow of the network models corresponding to the four economic modes. For specific coding, please refer to Supporting information S1 Code (a)–(d). For the flow of (a) the market economy on the far left, calculations can be made by following the branch shown by the dotted lines for (b) power, (c) gift, and (d) concession economies. In the initial setting, the number of nodes $n$, the mean degree $k_\mu$, the maximum number of steps $t_{max}$, and the graph $g(1)$ at $t = 1$ are set. $g(1)$ is a graph with no edges (the adjacency matrix is zero). At step $t$ in the calculation flow, for each of the four economic modes, the selection from the node list, the rewriting of the node list (in the case of (a) market and (b) power economies), and the generation of edges, the deletion of edges (if the mean degree exceeds $k_\mu$), and the drawing of the graph $g(t)$ are sequentially performed. These steps are repeated until the number of steps reaches $t_{max}$.

Here, we compare the simple random graph model with two representative random graph models that are structurally nuanced models: the Watts–Strogatz (WS) model and the Barabási–Albert (BA) model.

The WS model is formed by first constructing an undirected regular graph with $n$ nodes, each having a predetermined degree (the number of nodes adjacent to each node is equal), and then randomly reconnecting the undirected edges of

**Table 1. Four economy modes and setting of network models.**

|  | (a) Market economy | (b) Power economy | (c) Gift economy | (d) Concession economy |
|---|---|---|---|---|
| Polanyi | Market exchange | Redistribution | Reciprocity | – |
| Karatani | Commodity exchange | Plunder and redistribution | Reciprocity | Recovery of reciprocity in a higher dimension |
| Graeber | Exchange | Hierarchy | Baseline communism |  |
| Deguchi | – | – | – | We-turn |
| Node list | Selected-nodes priority (enclosure) | Selected + all nodes (mitigation of enclosure) | All nodes (equality) | All nodes (equality) |
| Edge direction | Bidirection (exchange) | Bidirection (exchange) | Bidirection (obligation to return) | Unidirection (de-obligation) |
| Edge generation | Random | Random | Random | Random |
| Edge deletion | Random (exchange failure) | Random (exchange failure) | Random (return failure) | Time passage (oblivion) |

this graph with a rewiring probability [27]. Here, we use a primitive method to convert undirected edges to bidirectional edges and rewire bidirectional edges as a set, and a derivative method to rewire one of the bidirectional edges (i.e., unidirectional edge). See S1 Code (e) and (f).

The conventional BA model is formed by first building a complete graph (where all nodes are connected to each other) with a small number of nodes, and then adding nodes of a specified degree until a specified number of nodes is reached [28]. Here, for comparison, we form a BA model network by adding bidirectional edges to a graph $g(1)$ with node number $n$ in the same way as economic modes (a) to (c). However, these edges are randomly connected with a probability proportional to the degree of the nodes, as in the Barabási–Albert method. See S1 Code (g).

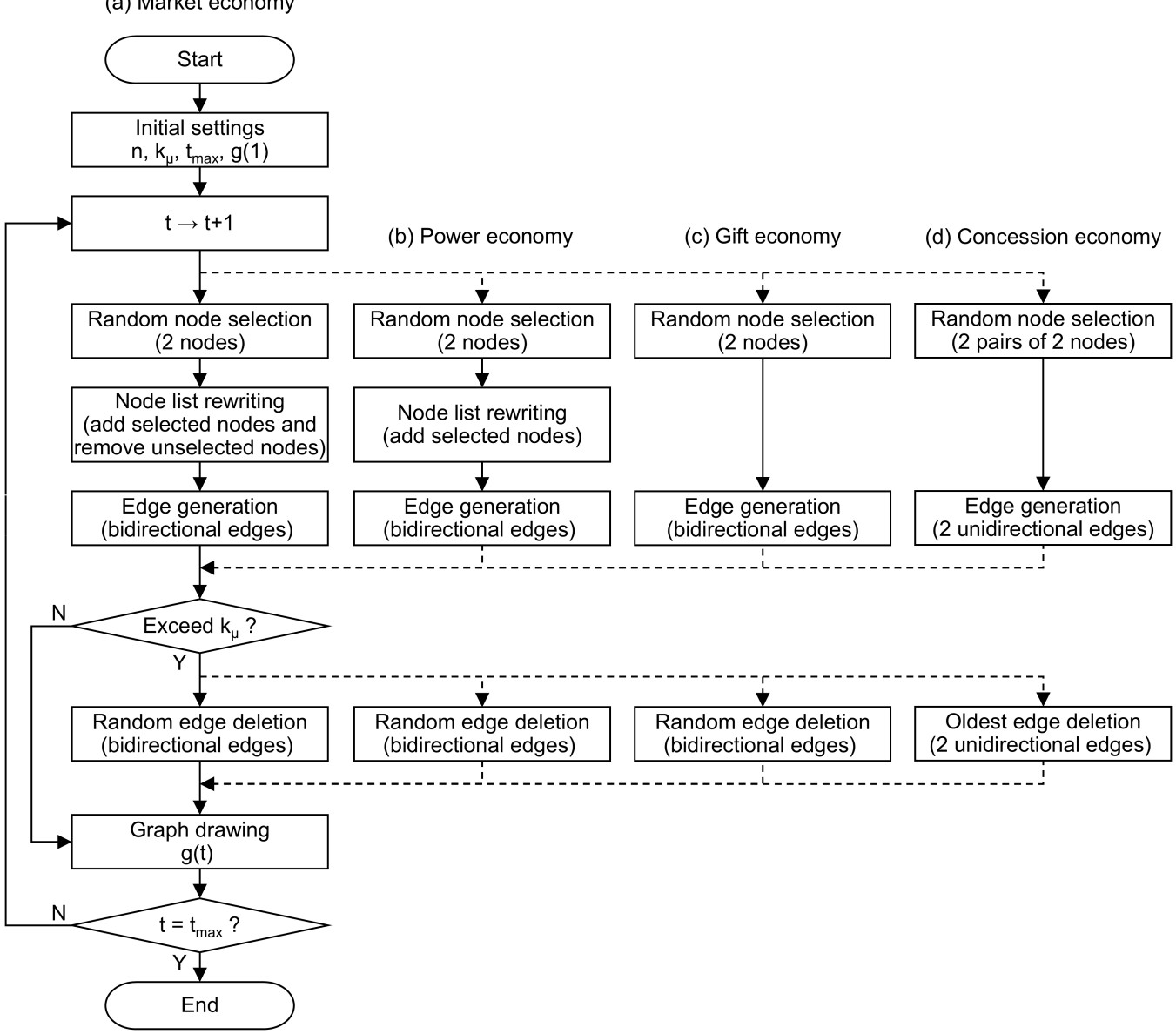

**Fig 2. Calculation flow of four-mode network models.**

## Network features

We focus on the cluster coefficient, graph density, reciprocity, assortativity, and centrality as features to measure the properties of the four economic modes. For the calculation methods of typical features, please refer to references [30,31]. These features are calculated for the graph $g(t)$ at each step.

The cluster coefficient is a feature that indicates the degree of clustering between nodes and is calculated as the ratio of the actual number of links to the possible number of links between neighboring nodes. The local clustering coefficient $C_i$ for node $i$ is expressed in Eq (1), using the degree $k_i$ of node $i$ and the number of actual links $e_i$ of the neighboring nodes. The mean cluster coefficient $C$ for the entire graph is the mean of the local cluster coefficient $C_i$ for the number of nodes $n$, and is expressed as in Eq (2).

$$C_i = \frac{2e_i}{k_i(k_i-1)}$$
(1)

$$C = \frac{1}{n}\sum_i^n C_i$$
(2)

Graph density is a feature that indicates the density of the edges connecting nodes and is calculated as the ratio of the number of actual edges to the number of possible edges. The density $D$ of a directed graph is expressed as in Eq (3), using the number of nodes $n$ and the number of actual edges $m$.

$$D = \frac{m}{n(n-1)}$$
(3)

Reciprocity is a feature that indicates the mutual relationship between edges and is calculated as the ratio of the number of bidirectional edges to the number of edges in the graph. The reciprocity $R$ is expressed as in Eq (4), using the number of edges $m$ and the number of bidirectional edges $m_r$.

$$R = \frac{m_r}{m}$$
(4)

Assortativity is a feature that indicates the connectivity from one node to another similar node and is calculated based on the correlation coefficient of the degrees between two nodes. Specifically, the assortativity $S$ is expressed as in Eq (5), using the element $a_{ij}$ of the adjacency matrix $A$ for nodes $i$ and $j$, the out-degree $d_i$ and $d_j$, the number of edges $m$, and the presence/absence $\delta_{ij}$ (0 or 1) of an edge between $i$ and $j$.

$$S = \frac{\sum_{i,j}^n \left(a_{ij} - \frac{d_i d_j}{2m}\right)}{\sum_{i,j}^n \left(d_i \delta_{ij} - \frac{d_i d_j}{2m}\right)}$$
(5)

Centrality is a feature that indicates the importance of a node in the network, and closeness centrality is calculated from the average distance to all other nodes connected to the node. The closeness centrality $c_i$ of node $i$ is expressed as in Eq (6), using the average distance $l_i$, the number of nodes $n_c$ connected to node $i$, and the element $d_{ij}$ of the distance matrix between $i$ and $j$. The mean $c_\mu$ and deviation $c_\sigma$ of the closeness centrality for all nodes are expressed as Eqs (7) and (8), respectively.

$$c_i = \frac{1}{l_i} = \frac{n_c}{\sum_j^{n_c} d_{ij}}$$
(6)

$$c_\mu = \frac{\sum_i^n c_i}{n} \tag{7}$$

$$c_\sigma = \sqrt{\frac{1}{n}\sum_i^n (c_i - c_\mu)^2} \tag{8}$$

Unlike the features described above, the Gini coefficient is not calculated for the graph $g(t)$ for each step but rather for a graph with sufficient edges generated, such as $g(t_{max})$. The Gini coefficient is calculated based on the distribution of wealth given to a node and propagated to other nodes.

Using the element $a_{ij}$ of the adjacent matrix $\boldsymbol{A}$ of $g(t_{max})$, the element $p_{ij}$ of the propagation matrix $\boldsymbol{P}$ is expressed as in Eq (9). This equation implies that the wealth is equally distributed among the edges from nodes $i$ to h.

$$p_{ij} = \frac{a_{ij}}{\sum_h^n a_{hj}} \tag{9}$$

Randomly selecting a node and assigning it a wealth of 1 is represented as an $n$-row column vector $\boldsymbol{v}_0$, with only one element being 1 and the rest being 0. By repeating the product of the propagation matrix $\boldsymbol{P}$ with $\boldsymbol{v}_0$ for a number of steps $\tau$, the column vector $\boldsymbol{v}_\tau$ after the propagation of wealth can be obtained as in Eq (10). The elements of $\boldsymbol{v}_\tau$ represent the distribution of wealth for all nodes.

$$\boldsymbol{v}_\tau = \boldsymbol{P}^\tau \cdot \boldsymbol{v}_0 \tag{10}$$

The Gini coefficient is a well-known index for evaluating the degree of inequality in the distribution of wealth [32] and is calculated by drawing the Lorenz curve and the equal distribution line [33]. Various indices can be calculated from the Lorenz curve; however, we used the Gini coefficient, which is the most common.

By operating on Eq (11), the elements $v_{\tau i}$ of $\boldsymbol{v}_\tau$ at step $\tau$ are sorted from smallest to largest, and the $j$-th wealth from the smallest is taken as $u_{\tau j}$, and the Gini coefficient $G$ is calculated using Eq (12). If wealth is distributed evenly, $G$ is 0; if in a delta distribution (all wealth is concentrated in just one node), $G$ is 1. In other words, the greater the inequality, the larger the Gini coefficient. The Gini coefficient for $\boldsymbol{v}_0$ is 1.

$$u_{\tau j} = Sort_j(v_{\tau i}) \tag{11}$$

$$G = \frac{2 \cdot \sum_j^n j \cdot u_{\tau j}}{n \sum_j^n u_{\tau j}} - \frac{n+1}{n} \tag{12}$$

For the simple random graph model of the four economic modes, the clustering coefficient $C$, graph density $D$, reciprocity $R$, assortativity $S$, mean centrality $c_\mu$ and deviation $c_\sigma$ are calculated for graph $g(t)$ at each step, and the Gini coefficient $G$ is calculated using $g(t_{max})$. For the WS model and BA model, which are the subjects for comparing the four modes, the network features are calculated for the finally formed network using the same method as above.

## Results

### Four economic modes

First, we set the parameters that are common to four-mode network models. The number of nodes, that is, the number of people in the community, was set to $n = 100$, based on Dunbar's number (the maximum number of people in a group that can maintain a stable social state) [34,35]. The mean degree was set to a relatively small value, $k_\mu = 4$, based on the degree

of social networks [36]. If the mean degree (or total degree = number of nodes × mean degree) is too large, all models will approach a complete graph, making it difficult to observe differences in their properties. In the calculation flow shown in Fig 2, two edges are generated per step, and the upper limit of the total degree ($n \cdot k_\mu = 400$) is reached at the 200-th step. Therefore, the maximum number of steps is set to $t_{max} = 600$, sufficiently larger than 200. See S1 Code (a) to (d).

Fig 3A1 shows the results of the network graph of (a) the market economy at step $t = 80$. Similarly, 3B1 shows (b) the power economy, 3C1 shows (c) the gift economy, and 3D1 shows (d) the concession economy. Fig 3A2 shows (a) the market economy at step $t = 400$, 3B2 shows (b) the power economy, 3C2 shows (c) the gift economy, and 3D2 shows (d) the concession economy. The spring-electrical embedding method (a method that minimizes energy by setting the edges as springs and the nodes as electric charges) was used for the drawing.

In Fig 3, the network gradually expands from (a) the market economy (A1, A2) to (d) the concession economy (D1, D2). In (a), the market economy, the network is concentrated in some nodes because priority is given to previously selected nodes to simulate the enclosure. In (c), the gift economy (C1, C2), all nodes were treated equally, which leads the network to expand, and in (b), the power economy (B1, B2), because the enclosure is mitigated, the network becomes an intermediate between (a) the market economy and (c) the gift economy. In (d), in the concession economy (D1, D2), the network is further expanded because the nodes are not bound by bidirectional edges.

Fig 4A1 shows the calculation results of the mean cluster coefficient $C$ and graph density $D$ for (a) the market economy. The horizontal axis represents the number of steps $t$, and the vertical axis represents the mean cluster coefficient $C$ and graph density $D$. The results of the three calculations are shown by the three blue and three green lines, respectively. Similarly, 4B1 shows (b) the power economy, 4C1 shows (c) the gift economy, and 4D1 shows (d) the concession economy. Fig 4A2 shows the calculation results for reciprocity $R$ and assortativity $S$ for (a) the market economy. The horizontal axis represents the number of steps $t$, and the vertical axis represents reciprocity $R$ and assortativity $S$. The results of the three calculations are indicated by blue and green lines, respectively. Similarly, 4B2 shows (b) the power economy, 4C2 shows (c) the gift economy, and 4D2 shows (d) the concession economy.

In Fig 4, regarding the mean clustering coefficient $C$ (blue lines), (a) the market economy (A1) has the largest value and the smallest fluctuation. This is because the network is concentrated at some nodes owing to the enclosure and does not change significantly, even if the edges are deleted. In (b) the power economy (B1), the value is smaller than in (a) the market economy (A1) owing to the mitigation of the enclosure. In (c) the gift economy (C1) and (d) the concession economy (D1), the values increase as the steps progress but fluctuate dynamically around 0.04, owing to edge deletion.

Regarding the graph density $D$ (green lines), the value for (a) the market economy (A1) is roughly 0.02 as the steps progress, and the values for the other (b) power economy (B1), (c) gift economy (C1), and (d) concession economy (D1) are 0.04 at step $t = 200$. The value of 0.04 is determined from total degree 400 (= number of nodes $n \times$ mean degree $k_\mu$) and Eq (3). In (a), the market economy (A1), the value is smaller than 0.04 because the network is concentrated in some nodes.

Regarding reciprocity $R$ (blue lines), in (a) the market economy (A2), (b) the power economy (B2), and (c) the gift economy (C2), the values are always 1 because bidirectional edges (exchange or obligation to return) are generated. In contrast, in (d) the concession economy (D2), the value is small because unidirectional edges (the obligation to return) are randomly generated, and bidirectional edges are rarely generated by chance.

Regarding assortativity $S$ (green lines), the values are small in the (a) market economy (A2), (b) power economy (B2), (c) gift economy (C2), and (d) concession economy (D2). This is because the edges are randomly generated without relying on a specific probability distribution. The time-series fluctuation in the value of (a), the market economy (A2), is small because, as already mentioned, the network remains almost unchanged even when an edge is deleted.

Fig 5A1 shows the calculation results of the mean $c_\mu$ and deviation $c_\sigma$ of centrality for (a) market economy. The horizontal axis is the number of steps $t$, and the vertical axis is the mean $c_\mu$ and deviation $c_\sigma$. The results of the three calculations are shown by the three blue and three green lines, respectively. Similarly, 5B1 shows (b) the power economy, 5C1 shows (c) the gift economy, and 5D1 shows (d) the concession economy. Fig 5A2 shows the calculation results of

 

the Gini coefficient $G$ for (a) market economy. The horizontal axis represents the number of steps $\tau$, and the vertical axis represents the Gini coefficient $G$. The results of the three calculations are represented by three blue lines. Similarly, 5B2 shows (b) the power economy, 5C2 shows (c) the gift economy, and 5D2 shows (d) the concession economy.

In [Fig 5](), regarding the mean centrality $c_\mu$ (blue lines) and deviation $c_\sigma$ (green lines), in (a) market economy (A1), the mean $c_\mu$ value is smaller, and the deviation $c_\sigma$ value is larger than in the other economies. This is because some nodes are prioritized owing to the enclosure, and the difference between these and other nodes is large. In (b) the power

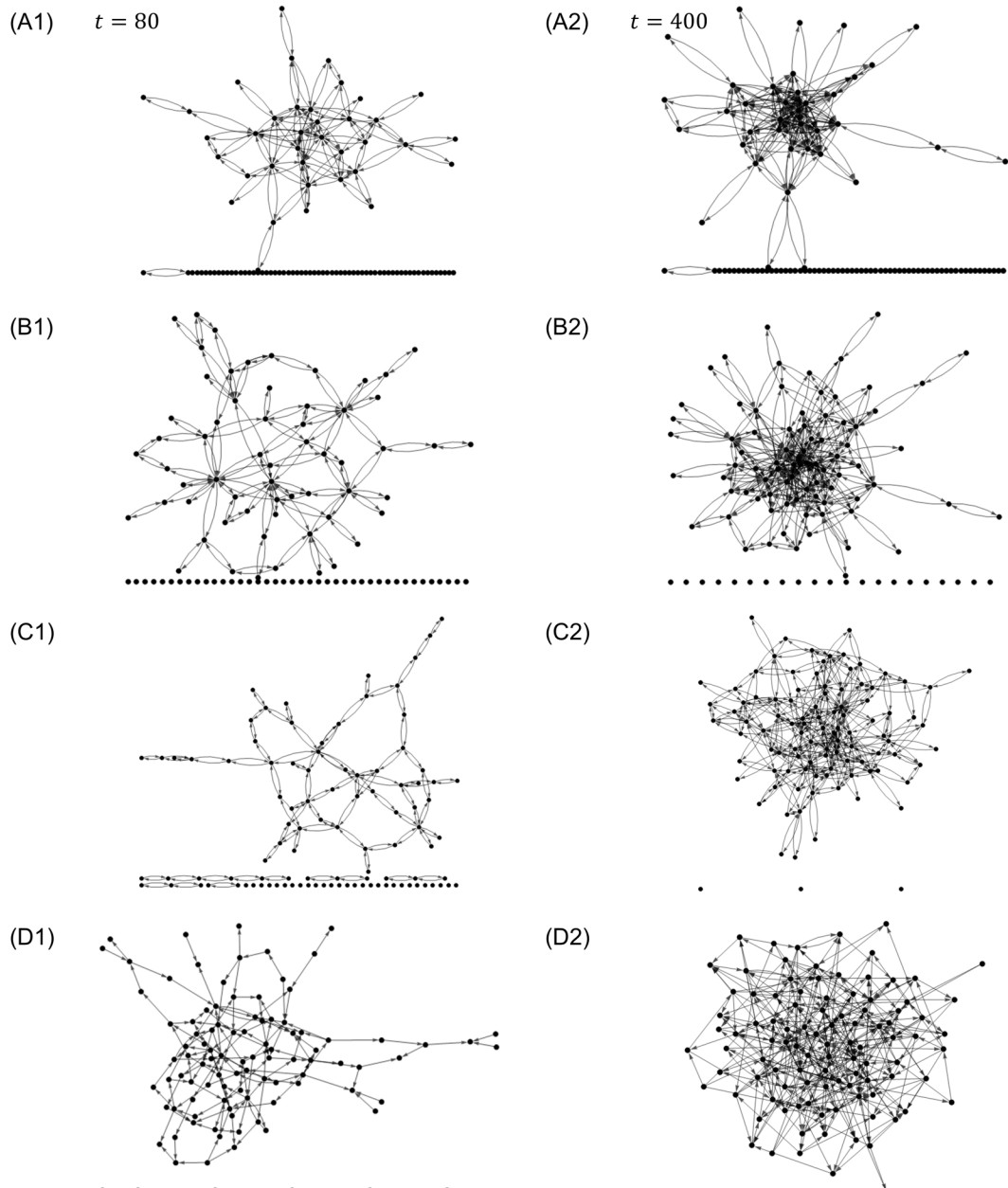

**Fig 3. Network graphs.** (A1) market, (B1) power, (C1) gift, and (D1) concession economies at $t = 80$. (A2) market, (B2) power, (C2) gift, and (D2) concession economies at $t = 400$.

economy (B1), the value of mean $c_\mu$ becomes larger, and the value of deviation $c_\sigma$ becomes smaller than in (a) market economy (A1) due to the mitigation of enclosure. In (c) the gift economy (C1) and (d) the concession economy (D1), all nodes are treated equally, and the value of the deviation $c_\sigma$ becomes even smaller. Similarly to the graph density $D$ shown in Fig 4, the values of the mean $c_\mu$ and deviation $c_\sigma$ are approximately constant at $t = 200$, where the total degree is limited by the mean degree $k_\mu$. The values of the mean $c_\mu$ and deviation $c_\sigma$ are stable in contrast to the dynamic fluctuations in the mean cluster coefficient $C$ in Fig 4, and it is considered that the equality of the nodes is maintained.

Regarding the Gini coefficient $G$ (blue lines), (a) market economy (A2) has the highest value, and as the steps progress, the value decreases to 0.8. Wealth is propagated only between some nodes because of the enclosure. In (b) power economy (B2), the value decreases to 0.5 as the steps progress, owing to the mitigation of the enclosure. Incidentally, the global Gini coefficient has remained high at 0.7 over the years [1] and exceeded the threshold of 0.4 of the warning level for social unrest [37]. The values for (a) the market economy (A2) and (b) the power economy (B2) reflect the global situation. In contrast, in (c) the gift economy (C2) and (d) the concession economy (D2), all nodes are treated equally, and wealth is distributed; therefore, the value drops to 0.4 or less. This value represents the level of economic health and social stability [37]. Additionally, when comparing (c) gift economy and (d) concession economy, the changes in (d) concession economy are more rapid. This is because (d) the concession economy has unidirectional edges; wealth stagnates in the early stages of the step, but as the steps progress, wealth propagates simultaneously.

## Four modes with WS and BA models

In the WS model, to align the number of nodes $n = 100$ and the mean degree $k_\mu = 4$ as the same as the four modes, we first form an undirected regular graph with the number of nodes $n = 100$ and the degree $k_{WS} = 2$. The rewiring probability $\rho$ is set to 0.7 based on survey results for various real networks [36]. Then, we randomly reconnect the undirected edges of the regular graph with the rewiring probability $\rho = 0.7$, and convert the undirected edges into bidirectional directed edges. In this model, the bidirectionality of edges is always maintained. See S1 Code (e).

As a derivative of the WS model (WS'), we first form a regular graph with node number $n = 100$ and degree $k_{WS} = 2$, treat undirected edges as bidirectional directed edges, and randomly reconnect each directed edge with the rewiring probability $\rho = 0.7$. In this derivative WS' model, rewired unidirectional edges and unrewired bidirectional edges coexist. See S1 Code (f).

Regarding the BA model, it is necessary to align the total number of nodes $n = 100$ and the total degree $n \cdot k_\mu = 400$ as the same as the four modes. Therefore, as in economic modes (a) to (c), bidirectional edges are added to a graph with $n = 100$ nodes. Here, edges are randomly generated with a probability proportional to the degree of the node, based on the conventional BA model approach. See S1 Code (g).

Figs 6 and 7 show box-and-whisker plots of the statistics of a hundred calculations of network features for (a) market, (b) power, (c) gift, and (d) concession economies, WS, WS', and BA models. For (a)–(d), network features have been calculated for $g(t_{max})$, and for the WS, WS', and BA models, network features have been calculated for the final-formed network graphs. Fig 6A shows mean clustering coefficient $C$, 6B shows graph density $D$, 6C shows graph reciprocity $R$, 6D shows graph assortativity $S$, 6E shows mean $c_\mu$ of closeness centrality, 6F shows its deviation $c_\sigma$, and Fig 7 shows Gini coefficient $G$ ($\tau = 100$). Refer to S1 Data (a)–(g) for the calculation results.

In Fig 6A, the mean clustering coefficient $C$ decreases gradually from (a) market economy to (b) power, (c) gift, and (d) concession economies. This is because in (a) market economies, enclosure generates large, fixed clusters, in (b) a power economy, enclosure is relaxed, and in (c) gift and (d) concession economies, enclosure does not occur. The value of the WS model, which rewires bidirectional edges, is the smallest, while the value of the WS' model, which rewires unidirectional edges, is larger than the WS model. The value of the BA model is about the same as that of (b) a power economy.

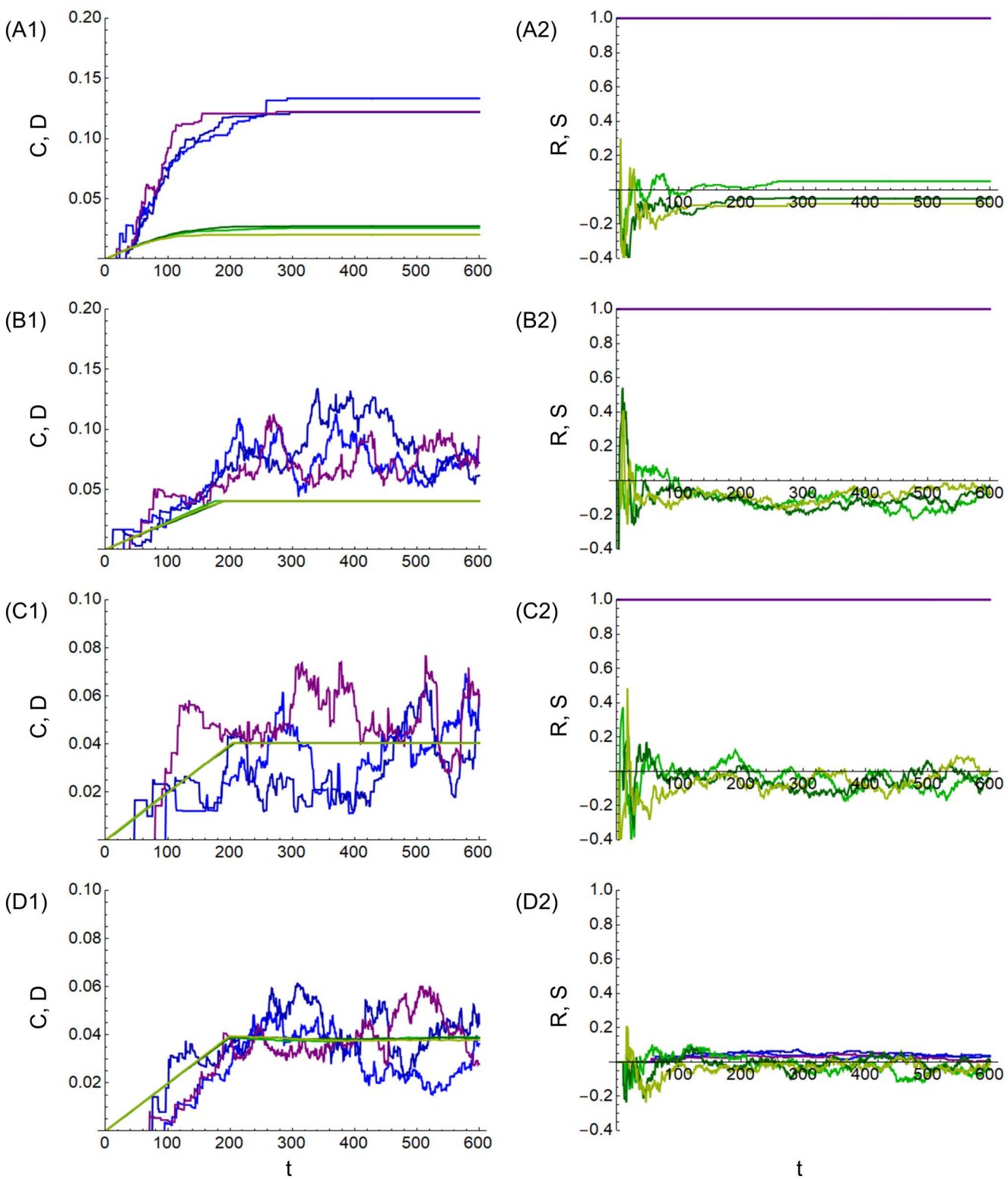

**Fig 4. Calculation results of mean clustering coefficient, graph density, graph reciprocity, and graph assortativity.** (A1) market, (B1) power, (C1) gift, and (D1) concession economies for mean clustering coefficient $C$ (blue lines) and graph density $D$ (green lines). (A2) market, (B2) power, (C2) gift, and (D2) concession economies for graph reciprocity $R$ (blue lines) and graph assortativity $S$ (green lines).

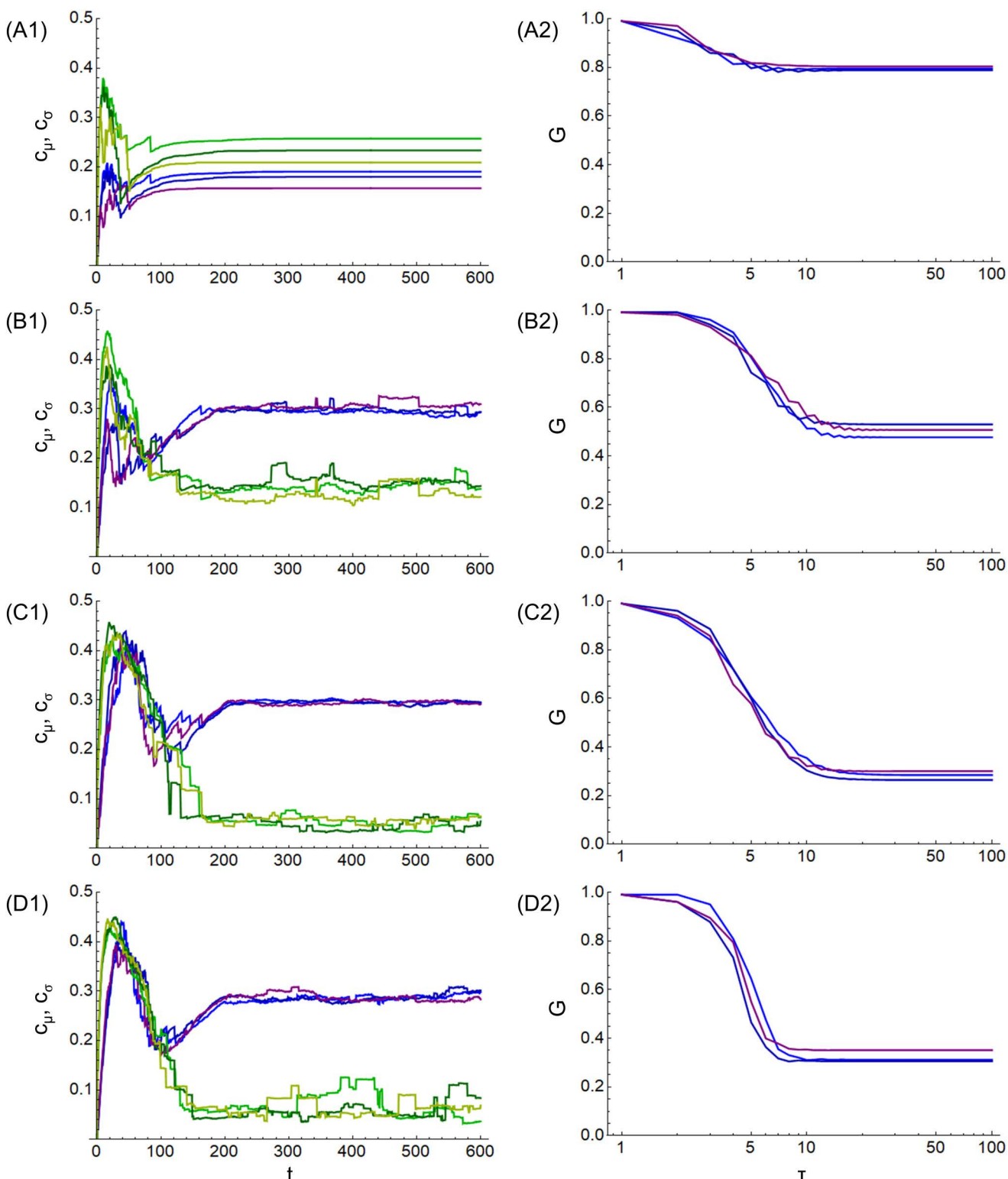

**Fig 5. Calculation results of closeness centrality and Gini coefficient.** (A1) market, (B1) power, (C1) gift, and (D1) concession economies for mean $c_\mu$ (blue lines) and deviation $c_\sigma$ (green lines) of closeness centrality. (A2) market, (B2) power, (C2) gift, and (D2) concession economies for Gini coefficient $G$ (blue lines).

In Fig 6B, except for (a) market economy, the graph density $D$ is almost the same value of 0.04, which is determined by the number of nodes and the mean degree. The value of (a) market economy differs from the others because, as mentioned in Fig 4, the network is concentrated on specific nodes due to enclosure.

In Fig 6C, the reciprocity $R$ is 1 in (a) market, (b) power, and (c) gift economies, as well as in the WS and BA models, which consist of bidirectional edges. In (d) a concession economy, unidirectional edges are generated randomly, so the

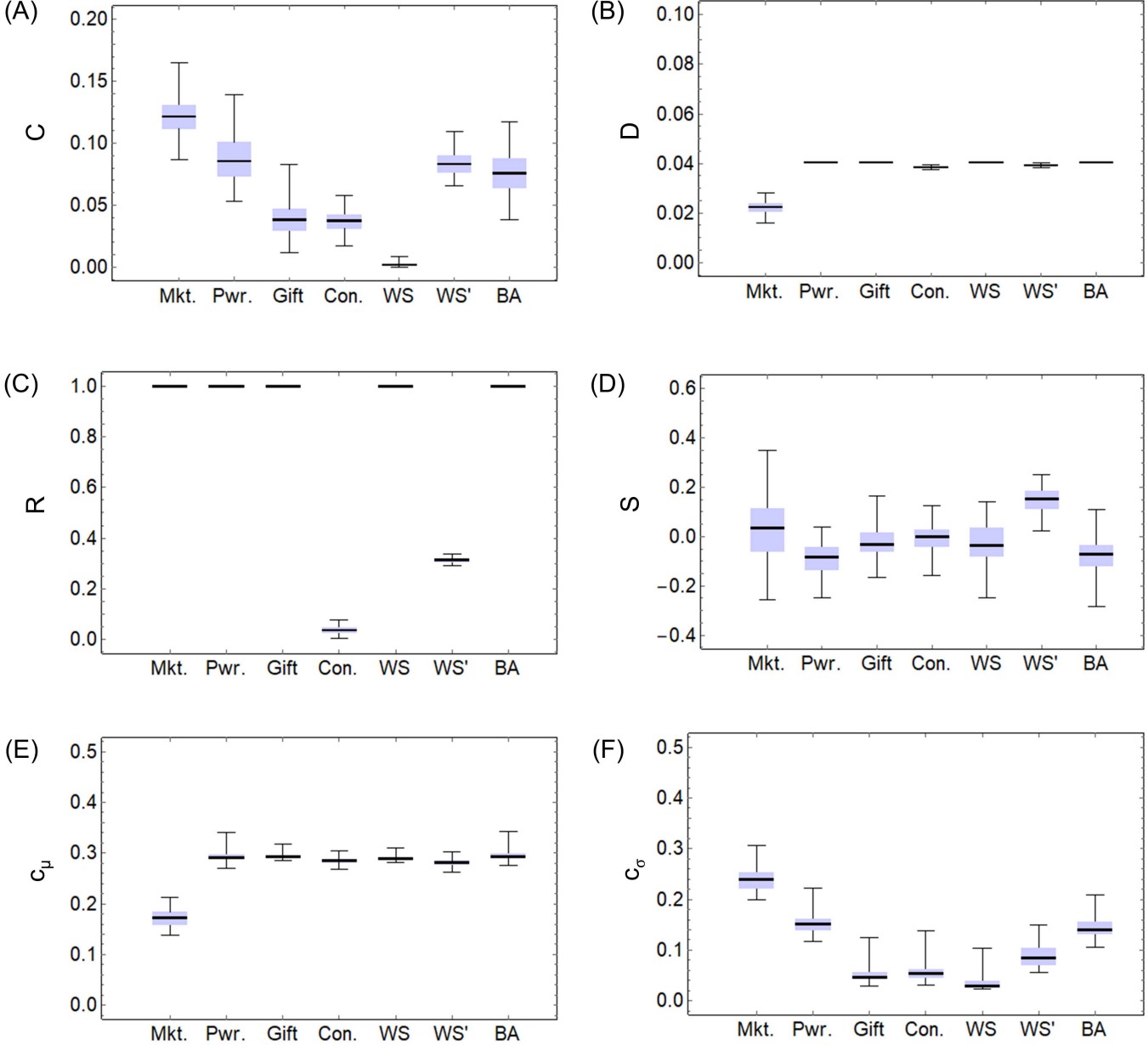

**Fig 6. Box-and-whisker plots of network features.** (A) mean clustering coefficient $C$, (B) graph density $D$, (C) graph reciprocity $R$, (D) graph assortativity $S$, (E) mean $c_\mu$ of closeness centrality, and (F) its deviation $c_\sigma$ for market, power, gift, and concession economies, WS, WS', and BA models.

value is close to 0. In the WS' model, some bidirectional edges remain even after rewiring, so the value is roughly 0.3 ($= 1 - \rho$).

In Fig 6D, there is little difference in the values of assortativity $S$, but the values of the WS' model are slightly larger. The reason for the large variance in (a) market economy is thought to be the large differences in network shapes due to enclosure with each calculation.

In Fig 6E, except for (a) market economy, the values of centrality $c_\mu$ are almost the same at 0.3. The (a) market economy value differs from others because, as mentioned in Fig 5, specific nodes are prioritized by enclosure, resulting in large differences between them and other nodes.

In Fig 6F, the centrality deviation $c_\sigma$ gradually decreases from (a) a market economy to (b) power, (c) gift, and (d) concession economies. This is because there is a difference in centrality between specific nodes and other nodes in (a) market and (b) power economies. The value of the WS model is the smallest, while the value of the BA model is the same as in (b) the power economy.

In Fig 7, the Gini coefficient $G$ gradually decreases from (a) a market economy to (b) power and (c) gift economies. Although the reciprocity $S$ of (d) the concession economy is 0, its Gini coefficient $G$ is less than the social unrest warning level of 0.4 [37]. The value of the WS model is the smallest, while the value of the WS' model is larger than that of (d) the concession economy. The value of the BA model, as well as other network features, is similar to (b) the power economy's. This shows that the situation in which the nodes are selected in proportion to their degree in the BA model is generally equivalent to the situation in which the selected nodes are cumulatively more likely to be selected in (b) the power economy. In other words, (b) a power economy will have a skewed distribution of wealth (power law) like the BA model.

Comparing Fig 7 with Fig 6, the trend of the Gini coefficient $G$ is very similar to that of the cluster coefficient $C$ and the centrality deviation $c_\sigma$. Fig 8 shows scatter plots of the cluster coefficient $C$ and centrality deviation $c_\sigma$ for the Gini coefficient $G$.

In Fig 8, the correlation coefficients $r$ of the cluster coefficient $C$ and the centrality deviation $c_\sigma$ are 0.92 and 0.97, and the coefficients of determination $R^2$ are 0.98 and 0.96, respectively, indicating that both are strongly correlated with the Gini coefficient $G$. To reduce the Gini coefficient (i.e., to suppress wealth inequality), it is necessary to avoid generating fixed clusters and central nodes (i.e., to open up enclosure and share wealth equally among all members). These are connected to the softness and hollowness of the We-turn philosophy.

Fig 9 shows the results of calculating the Gini coefficient $G$ ($\tau = 100$) for the WS and WS' models with the rewiring probability $\rho$ as a parameter. The blue plots and lines are the WS model, the green plots and lines are the WS' model, and the whiskers on the plots indicate the standard deviation. Refer to S1 Data (h)–(i) for the results of a hundred calculations.

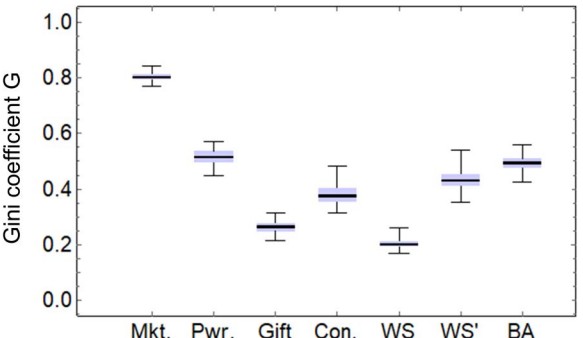

**Fig 7. Box-and-whisker plots of Gini coefficients.** Gini coefficients $G$ for market, power, gift, and concession economies, WS, WS', and BA models.

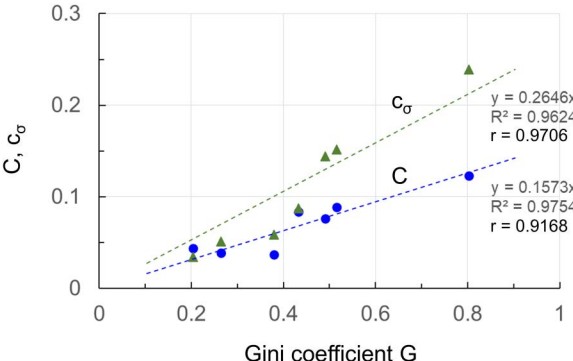

**Fig 8. Scatter plots of clustering coefficient and centrality deviation for Gini coefficient.** The x-axis is the Gini coefficient $G$, and the y-axis is the clustering coefficient $C$ and centrality deviation $c_\sigma$.

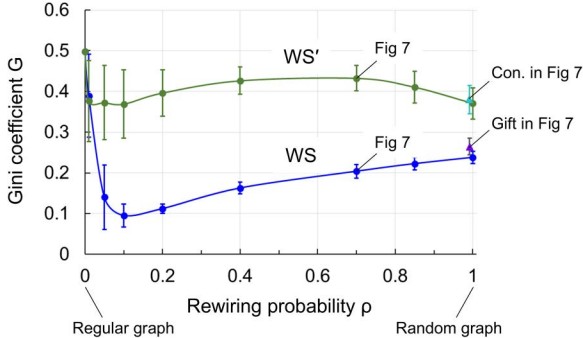

**Fig 9. Calculation results of Gini coefficient for WS and WS' models.** The horizontal axis is the rewiring probability $\rho$, the vertical axis is the Gini coefficient $G$, the blue and green plots are the WS and WS' models, respectively.

In Fig 9, the Gini coefficient $G$ of the WS model having full reciprocity is smaller than that of the WS' model lacking reciprocity. In the vicinity of the rewiring probability $\rho = 0.1$, bypass edges are formed for the initial regular graphs of the WS and WS' models, causing the Gini coefficient $G$ to decrease, and when $\rho$ exceeds 0.1, the regular graphs collapse more widely, causing the Gini coefficient $G$ to gradually increase.

The results for (c) the gift economy and (d) the concession economy in Fig 7 correspond to the results for the WS and WS' models in Fig 9 with a rewiring probability of $\rho = 1$, respectively. The random wiring and orderly oblivion of (d) the concession economy are generally equivalent to the random rewiring of the WS' model. The results for the WS and WS' models in Fig 7 correspond to the results of $\rho = 0.7$ in Fig 9. From another perspective, the variations of (c) gift and (d) concession economies can be considered within the spectrum created by the WS and WS' models. As shown in Fig 7, the (d) concession economy has a smaller Gini coefficient $G$ than the (a) market and (b) power economies. However, to further reduce the Gini coefficient, it would be effective to approach the WS model in Fig 9, that is, to form a small-world community by fostering natural reciprocity while avoiding the constraints of the obligation to return in the (c) gift economy.

For reference, in relation to our study which is based on a network model, other research is derived from an econophysics agent-based model [38]. This research compares a combined model of equivalent exchange and redistribution (EX model) with a non-equivalent exchange model (NX model). The EX model corresponds to (a) the market economy exchange and (b) power economy redistribution in this study, while the non-equivalent exchange in the NX model is positioned between the

bidirectional (c) gift economy and the unidirectional (d) concession economy (between WS and WS' in Fig 9). Although the calculation results of both models regarding the relationship between economic flows and the Gini coefficient are similar, it is concluded that the NX model (mutual aid) is "qualitatively superior" to the EX model (compulsory and frequent taxation) in reducing inequality and stimulating economic flows. This conclusion supports the results of this study.

## Discussion

The calculation results of the network model corresponding to the four economic modes reveal that, first, inequality is suppressed in the following order: (d) concession economy, (c) gift economy, (b) power economy, and (a) market economy; second, enclosure in (a) a market economy is the main cause of inequality, while redistribution in (b) a power economy contributes to mitigating enclosure; third, (c) gift and (d) concession economies bring about healthy and equal societies; and fourth, the original question of whether (d) concession economy can be established even without the guarantee of exchange or reciprocity can be resolved. This is the first time, from the perspective of a mathematical network model, that the recovery of reciprocity in a higher dimension by Karatani, the baseline communism of Graeber, and the We-turn of Deguchi are economically possible.

When comparing the simple random graph model used to calculate the four economic modes with structurally meaningful random graph models, the following findings were obtained: (b) the power economy is generally equivalent to the BA model, which produces scale-freeness and a power law distribution of wealth; (c) the gift economy and (d) the concession economy correspond to the WS and WS' models with the rewiring probability as 1, respectively; the variations of (c) the gift economy and (d) the concession economy are positioned within the parameters of the WS and WS' models' rewiring probability, and both economies are mutually transitional; the results of combining the four modes and WS, WS', and BA models show that the Gini coefficient is strongly correlated with the cluster coefficient and centrality deviation.

The small values of the centrality deviation and Gini coefficient in the (c) gift and (d) concession economies indicate that the community is equal or hollow (i.e., there is no specific authority). Furthermore, it was found that in a community, maintaining large clusters is not necessarily better, such as in (a) market and (b) power economies. However, the continued dynamic formation of moderate clusters leads to a healthier and more equal economy, such as in (c) gift and (d) concession economies.

The (d) concession economy is preferable to (c) a gift economy in that it is free from the constraints of obligation to return; conversely, the key is how to continue to circulate concessions while retaining the de-rule and de-obligation in the absence of guaranteed reciprocity. From the perspective of (a) the market economy, it is essential to mitigate the enclosure of the commons and recover community relationships. To put this into practice, evoking baseline communism and increasing We-turn is necessary.

(d) Concession economy: More specifically, it is necessary to eliminate the enclosure in capitalism–that is, the division between the wealthy and poor–and share the means of production, funds, and research as concessions, thereby promoting community cooperation. To encourage and continue circulating concessions, using cooperative platforms [39,40], for example, is beneficial. With the support of information technology, building a network of security and trust is possible to connect each person's contribution and reception as baseline communism and to compensate for "fundamental incapability" as "We" even without direct reciprocal relationships. This information technology would also include encouraging concessions to reduce the uneven distribution of wealth and warning against free riders who enjoy wealth unilaterally.

Additionally, several studies provide historical evidence to support our results. Research by anthropologist Kohler et al. [41] has shown that the Gini coefficient increases as society type shifts from hunting and gathering to horticulture and agriculture (from 0.17 to 0.37) and as political scale expands from family to locality, big man, region, and nation (from 0.15 to 0.38). This is because the domestication of large mammals, the expansion of farmland, and the development of cavalry warfare promoted the enclosure of the production means and the accumulation of wealth, thereby increasing inequality. According to research by anthropologist Smith et al. [42], hunter-gatherers are egalitarian in terms of opportunity and

outcome, which in this study would correspond to (c) a gift economy without enclosure or (d) a concession economy that does not demand equivalent returns.

Furthermore, research by economist Bowls et al. [43] has shown that inequality among hunter-gatherers and farmers in the Neolithic period was limited (0.362), that the introduction of the ox-drawn plows in the late Neolithic period increased land scarcity and labor redundancy (generating a dependent working class), and that the concentration of elite power in early primitive states continued to increase wealth inequality (0.695). Moreover, the authors touch on the similarities between prehistoric ox-drawn plows and modern AI and robots, arguing that enclosure of technology and concentration of power (in this study, (a) market and (b) power economies) expand wealth inequality.

Ours is the first study to present the feasibility of a concession economy from the network model perspective. Note that this study is a mathematical evaluation based on a primitive model and remains to be verified in the real world. Future challenges include the development of a cooperative platform and empirical research through fieldwork to implement a concession economy. In parallel with these efforts, we would like to promote the transformation from a capitalist economy to a concession economy through social activities that spread baseline communism and a We-turn to society.

## Supporting information

**S1 Code. Calculation codes of network models and features.**
(PDF)

**S1 Data. Calculation results data.**
(PDF)

## Acknowledgments

Professor Deguchi of Kyoto University and the Kyoto Institute of Philosophy taught us about the We-turn philosophy, while Emeritus Professor Hiroi of Kyoto University taught us about community economics and economic modes. We express our gratitude to them.

## Author contributions

**Conceptualization:** Takeshi Kato.

**Data curation:** Takeshi Kato.

**Methodology:** Takeshi Kato.

**Project administration:** Ryuji Mine.

**Software:** Takeshi Kato.

**Supervision:** Ryuji Mine.

**Visualization:** Takeshi Kato.

**Writing – original draft:** Takeshi Kato.

**Writing – review & editing:** Takeshi Kato, Junichi Miyakoshi, Misa Owa, Ryuji Mine.

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
