## [Decision Letter · Decision Letter 0]

12 May 2025

PONE-D-25-19624Market, power, gift, and concession economies: Comparison using four-mode primitive network modelsPLOS ONE

Dear Dr. Kato,

Thank you for submitting your manuscript to PLOS ONE. After careful consideration, we feel that it has merit but does not fully meet PLOS ONE’s publication criteria as it currently stands. Therefore, we invite you to submit a revised version of the manuscript that addresses the points raised during the review process.

We look forward to receiving your revised manuscript.

Kind regards,

Academic Editor

PLOS ONE

Journal Requirements:

Reviewers' comments:

Reviewer's Responses to Questions

**Comments to the Author**

1. Is the manuscript technically sound, and do the data support the conclusions?

Reviewer #1: Yes

Reviewer #2: No

2. Has the statistical analysis been performed appropriately and rigorously? 

Reviewer #1: Yes

Reviewer #2: No

3. Have the authors made all data underlying the findings in their manuscript fully available?

Reviewer #1: Yes

Reviewer #2: Yes

4. Is the manuscript presented in an intelligible fashion and written in standard English?

Reviewer #1: Yes

Reviewer #2: Yes

5. Review Comments to the Author

Reviewer #1: This manuscript proposes a novel comparison of four economic modes—market, power, gift, and concession—using network models inspired by the theories of Polanyi, Karatani, and Graeber. The topic is original and interdisciplinary, and the attempt to formalize economic systems via network metrics is commendable. However, the paper in its current form requires substantial revision before it can be considered for publication.

Major Issues:

1�Conceptual Clarity:

Key terms such as “concession economy” and “We-turn philosophy” lack clear definitions. Their theoretical grounding and practical implications should be elaborated. The relationships among the four economic modes (e.g., whether they are mutually exclusive, overlapping, or sequential) should also be clarified.

2�Modeling Details:

The paper does not provide enough information about how the network models are constructed. What are the assumptions, node/edge definitions, simulation parameters, or data sources? These need to be explained clearly.

3�Interpretation of Results:

The conclusions (e.g., “market economy leads to inequality,” “concession economy is ideal”) are too strong given the abstract nature of the models. The authors should avoid overgeneralization and provide more balanced, evidence-based discussion.

4�Language and Structure:

The manuscript requires careful language editing to improve clarity and flow. Some concepts are densely presented or ambiguously phrased.

Reviewer #2: The manuscript presents an interesting and highly conceptual framework grounded in philosophical theories such as “We-turn” and “baseline communism.” While the integration of these ideas into a network model is commendable, the manuscript lacks sufficient clarity and rigor in the definition of key modeling assumptions. Specifically, the mathematical encoding of core concepts such as "enclosure" and "de-obligation" within the network formation process is not clearly articulated. Additionally, the implementation of critical dynamics—such as edge deletion procedures (e.g., the “oblivion” mechanism in the concession economy)—remains vague. Greater technical transparency, including algorithmic steps or pseudocode, would significantly improve the model’s reproducibility.

The interdisciplinary ambition of the manuscript is appreciated, particularly its philosophical framing of economic modes. However, from a technical modeling perspective, the novelty is limited. The use of a simple random graph model (inspired by Kauffman’s button and thread model) does not substantially advance existing literature in network simulations. kindly refer:https://doi.org/10.1002/dac.6105, https://doi.org/10.1038/s41598-024-66515-x,
https://doi.org/10.1080/14786451.2024.2364226,
https://doi.org/10.1186/s44147-024-00460-4

https://doi.org/10.1016/j.icte.2024.05.002. https://doi.org/10.1016/j.heliyon.2024.e32217.

https://doi.org/10.1016/j.eij.2024.100577.,
https://doi.org/10.1002/dac.6105

https://doi.org/10.1002/eng2.13019. https://doi.org/10.18280/mmep.110801

https://doi.org/10.1002/dac.5901

The absence of comparison with more established and structurally nuanced models—such as Exponential Random Graph Models or agent-based models—reduces the strength of the manuscript’s methodological contribution.

The modeling approach relies on uniform edge generation probabilities across the different economic systems. This assumption may not adequately capture the structural and behavioral differences that the manuscript seeks to explore. While the use of metrics such as the Gini coefficient, assortativity, and centrality is appropriate, their application alone does not sufficiently support the manuscript’s broader philosophical and normative claims—particularly regarding the supposed feasibility or superiority of concession economies. Moreover, the model lacks sensitivity analysis or robustness checks to assess how outcomes vary with parameter changes.

The results are presented primarily through descriptive plots, without statistical validation or inferential analysis. To strengthen the manuscript's conclusions, the authors should consider including confidence intervals, standard deviations, or hypothesis tests. At present, some interpretations in the discussion—such as asserting the practical viability of the concession economy—appear speculative and are not adequately substantiated by the simulation results alone. The absence of detailed statistical summaries across multiple runs further limits the reliability of the findings.

While the authors describe the model as primitive, the study would benefit greatly from some form of empirical validation. Even a preliminary comparison with real-world data or stylized facts from socio-economic systems would improve the manuscript’s credibility. Without empirical grounding, it is difficult to assess whether the observed model dynamics have practical significance or are merely artifacts of the simulation assumptions.

6. PLOS authors have the option to publish the peer review history of their article (what does this mean? ). If published, this will include your full peer review and any attached files.

**Do you want your identity to be public for this peer review?** For information about this choice, including consent withdrawal, please see our Privacy Policy .

Reviewer #1: **Yes: ** Shilin Wang

Reviewer #2: No

---

## [Author Response · Author response to Decision Letter 1]

14 Jul 2025

We have upload a rebuttal letter as an attach file labeled 'Response to Reviewers.'

---

## [Decision Letter · Decision Letter 1]

29 Jul 2025

Market, power, gift, and concession economies: Comparison using four-mode primitive network models

PONE-D-25-19624R1

Dear Dr. Kato,

We’re pleased to inform you that your manuscript has been judged scientifically suitable for publication and will be formally accepted for publication once it meets all outstanding technical requirements.

Kind regards,

Md. Asaduzzaman, Ph.D., M. Engg.

Academic Editor

PLOS ONE

Additional Editor Comments (optional):

Reviewers' comments:

Reviewer's Responses to Questions

**Comments to the Author**

1. If the authors have adequately addressed your comments raised in a previous round of review and you feel that this manuscript is now acceptable for publication, you may indicate that here to bypass the “Comments to the Author” section, enter your conflict of interest statement in the “Confidential to Editor” section, and submit your "Accept" recommendation.

Reviewer #1: All comments have been addressed

Reviewer #2: All comments have been addressed

2. Is the manuscript technically sound, and do the data support the conclusions?

Reviewer #1: Yes

Reviewer #2: Yes

3. Has the statistical analysis been performed appropriately and rigorously? 

Reviewer #1: Yes

Reviewer #2: Yes

4. Have the authors made all data underlying the findings in their manuscript fully available?

Reviewer #1: Yes

Reviewer #2: Yes

5. Is the manuscript presented in an intelligible fashion and written in standard English?

Reviewer #1: Yes

Reviewer #2: Yes

6. Review Comments to the Author

Reviewer #1: The authors have carefully addressed all my previous comments and made the necessary revisions to the manuscript. The quality of the paper has been significantly improved, and I have no further concerns.

I recommend the manuscript for acceptance.

Reviewer #2: We kindly request that the article, now meeting all publication standards, be considered for release at the earliest convenience

7. PLOS authors have the option to publish the peer review history of their article (what does this mean? ). If published, this will include your full peer review and any attached files.

**Do you want your identity to be public for this peer review?** For information about this choice, including consent withdrawal, please see our Privacy Policy .

Reviewer #1: **Yes: ** Shilin Wang

Reviewer #2: No

---

## [Editor Report · Acceptance letter]

PONE-D-25-19624R1

PLOS ONE

Dear Dr. Kato,

I'm pleased to inform you that your manuscript has been deemed suitable for publication in PLOS ONE. Congratulations! Your manuscript is now being handed over to our production team.

Kind regards,

on behalf of

Dr. Md. Asaduzzaman

Academic Editor

PLOS ONE